# Upper-Limb Disturbances in Female Patients with Poland Syndrome, including the Digit Ratio (2D:4D)

**DOI:** 10.3390/jcm11247253

**Published:** 2022-12-07

**Authors:** Marta Fijałkowska, Mateusz Koziej, Bogusław Antoszewski

**Affiliations:** 1Department of Plastic, Reconstructive and Aesthetic Surgery, Medical University of Lodz, 90-419 Lodz, Poland; 2Department of Anatomy, Jagiellonian University Medical College, 31-008 Cracow, Poland

**Keywords:** Poland syndrome, digit ratio, upper limb

## Abstract

Background: Poland syndrome (PS) is a rare congenital anomaly characterized by a variable clinical picture. Classic deformity consists of the unilateral hypoplasia or aplasia of the pectoralis major muscle and ipsilateral hand malformations. The aim of this study is to present disturbances in the development of upper limb in women with Poland syndrome (including digit ratio 2D:4D) in comparison to the healthy controls. Methods: The group of patients with Poland syndrome consisted of 36 women, while the control group consisted of 50 heathy women. Both upper limbs were measured anthropometrically. The length of all fingers and forearms were measured, and the digit ratio was calculated. Results: In women with Poland syndrome, the length of digits 2 and 4 and the forearm were significantly higher on the nonaffected side than on the affected side. In addition, there were significant differences between the length of digits 2 and 4 and the forearm between patients and controls. Conclusions: In patients with Poland syndrome, the upper limb at the affected side is significantly different from the upper limb on the healthy side, mainly in the length of the forearm and digits. While examining the patient with Poland syndrome, we think it is essential to pay attention not only to hand anomalies but also to the development of whole upper limb. It may prove to be helpful in estimating the complete picture of Poland syndrome.

## 1. Introduction

Poland syndrome (PS) is a rare congenital anomaly characterized by a variable clinical picture. Classic deformity consists of the combination of the unilateral hypoplasia or aplasia of the sternocostal head of the pectoralis major muscle and ipsilateral hand malformations [1,2,3]. Moreover, on the affected side, hypoplasia or aplasia of the breast in women, the absence of axillary hair and abnormality in the axillary fold can be observed [4,5]. In addition, there are descriptions of Poland syndrome’s coexisting with anomalies of the skeletal system. Usually, it is demonstrated by the hypoplasia of costal cartilages and anterior portions of the ribs (especially the second through the fourth ribs), abnormalities of the sternum, protruding scapula, scoliosis, radioulnar synostosis, or subluxation of the elbow [4,5,6]. Poland syndrome can also be accompanied by mandibular prognathism and anomalies in the cranium, such as cranio-fronto-nasal dysplasia [6,7,8]. Some authors pay attention to the difference in the muscular system, namely the hypoplasia or aplasia of the remaining heads of the pectoralis major muscle and other muscles, specifically pectoralis minor, latissimus dorsi, trapezius, deltoid, serratus anterior, rectus abdominis, external oblique, intercostal muscles, infraspinatus, and supraspinatus [4,9,10,11].

The incidence of Poland syndrome reported by different authors varies, ranging from 1 in 17,000 to 1 in 100,000 births [3,4,5,6]. Poland syndrome is reported to be more common in men than in women [3]. An analysis of the gender and sidedness indicates that men are more likely to have a right-sided anomaly, whereas women have an approximately equal distribution of left- and right-sided anomalies [3,12].

In the literature, it is pointed out that hand deformities are not present in all patients with Poland syndrome, and that is why pectoral muscle deficiency is often used as the only defining point when diagnosis of Poland syndrome can be made [3,13]. The most common hand anomaly is partial or complete symbrachydactyly (absence of middle phalanges), but abnormalities may range from simple syndactyly to complete ectrodactyly (absence of digits) [3,14].

There are quite a lot of articles concerning hand malformations in Poland syndrome [4,6,14,15], although only few quite old reports present disturbances involving upper limbs [16,17,18], and to our knowledge, there are no reports describing 2D:4D ratios in hands in patients with Poland syndrome that compare those ratios to healthy controls.

## 2. Materials and Methods

The Department of Plastic, Reconstructive, and Aesthetic Surgery and our out-patient clinic has 54 patients with Poland syndrome under care. Anthropometric measurements of upper limbs were performed in 45 patients (42 women and 3 men). Because of the very small group of men, further analysis included only women. To be sure that limb growth had finished, the analysis was performed only on adults (36 women; the other 6 were below 18 years of age). The control group consisted of 50 healthy women at ages close to the examined group, and both upper limbs were measured (100 hands and forearms).

Both upper limbs were subjected to metric analysis. The following measurements were taken: the length of all digits and the length of the forearm (r-styr). Measurements of the digits were conducted on the palmar side of the hand by using a sliding caliper (GPM Instruments, Switzerland). During these measurements, the patient was sitting with her forearm and hand lying flat on the table with the internal surface of the palm upwards. The hand was entirely straight with the thumb lying a little bit laterally, and digits 2–5 were connected but without strong muscle tone. The points pseudophalangion (pph) and dactylion (da) are localized on the long axis of the digits: in proximal sulcus of the digit (pph) and the most distal on the pulp of the digit (da), respectively. The measurement of the forearm was taken between the points r-styr, where r (radiale) is the point on the top of the head of the radial bone and styr (stylion radiale) is the point on the top of the styloid process of the radial bone.

Quantitative features were usually described using the mean value ± standard deviation. Quantiles (Q1, Me, Q3) were applied if a nonparametric test had been performed to compare quantitative features in groups. Minimum and maximum values were added in both cases. Qualitative features were presented by frequencies and percentages. The Shapiro–Wilk test was applied to determine whether the quantitative data were normally distributed. To verify the homogeneity of variance, a Levene’s test was performed. To compare parameters between the left and right sides of patients with PS, the paired *t*-test or Wilcoxon test was used, depending on whether data were normally distributed. To compare the parameters of affected hand with the control group, the Mann–Whitney or *t*-test was used. The power analysis indicated that to detect a statistical difference of 2.5 mm between measured parameters (sigma = 6) by using a two-sided test, a 5% significance-level test (α = 0.05), and 80% power (β = 0.2), the required minimal sample size would need to be approximately 92. A *p*-value of <0.05 was considered to be statistically significant. The statistical analyses were performed with STATISTICA v13.1 (StatSoft Inc., Tulsa, OK, USA) for Windows.

## 3. Results

In total, 36 patients with Poland syndrome were examined, with a mean age of 24.4 ± 4.7 years (range 18 to 39 years). The women were divided into affected and nonaffected hand groups. The right side was affected in 13 cases (36.1%) (Figure 1), and the left side was affected in 23 cases (63.9%) (Figure 2). The control group consisted of 50 women, with a mean age of 24.3 ± 4.6 years (range from 18 to 38 years).

Measured parameters were compared for the affected side and the nonaffected side in patients with Poland syndrome. The results are presented in Table 1. The lengths of digits 2 and 4 were significantly higher on the nonaffected side (both *p* < 0.001). The r-styr parameter was statistically higher on the nonaffected side than on the affected side (*p* < 0.001). The mean of 2D:4D ratio was not significantly different between sides (*p* = 0.197). The analysis is presented in Figure 3.

Furthermore, the patients’ hand measurements were compared with the corresponding measurements of hands in the control group (Table 2). It revealed similar statistical dependences. There was a significant difference between the lengths digits 2 and 4 between groups, which was higher for the control group (both *p* < 0.05). The r-styr parameter was statistically higher in the control group (*p* < 0.001). The statistically significant difference of the 2D:4D ratio between groups was not shown (*p* = 0.208) (Figure 1).

Notable differences were observed between SD values of digit length, which were higher in the affected hands of patients with PS (SD = 12–13) than in the nonaffected hands in patients with PS and hands of the control group (ca. SD = 4.2 and SD = 2.5, respectively). This indicates that patients with Poland syndrome present greater variability in the size of their phalanges than that in healthy people.

## 4. Discussion

The etiology of Poland syndrome is still under discussion. Bavinck and Weaver believe that the cause of Poland syndrome is an interruption of the embryonic blood supply in the subclavian artery, the vertebral artery, and/or their branches, and these authors term this subclavian artery supply disruption syndrome (SASDS) [19]. The authors mention that possible mechanisms that may lead to the interruption of or a reduction in blood flow in these arteries are internal mechanical factors and external pressure on the blood vessels, such as an edema, a hemorrhage, a cervical rib, aberrant musculature, an amniotic band, intrauterine compression, or a tumor [19]. Additional evidence that seems to confirm vascular theory includes angiography studies. Galvagno et al. observed hypoplasia of the internal thoracic artery, and Beer et al. showed that missing the thoracoacromial artery and the thoracodorsal artery is a direct cause of the hypoplasia of pectoralis major muscle and latissimus dorsi muscle [20,21]. Other etiological theories include a disruption of the lateral mesodermal plate soon after fertilization and prenatal exposure to harmful social and physical environmental factors such as smoking and cocaine abuse, misoprostol intake, prenatal teratogens, and failed abortions [3,12,22,23,24]. Maybe there is also a link between limb development and exposure to hormones during fetal life, which is why we found it interesting to check the digit ratio 2D:4D in patients with Poland syndrome.

The relation between the lengths of the second digit and those of the fourth digit (2D:4D) proved to be a sexually dimorphic trait in humans [25]. Manning suggested that this ratio can be a marker of prenatal sex steroids exposure. He found that a low 2D:4D ratio is correlated with high fetal testosterone and low estrogens and that high 2D:4D is a result from low prenatal testosterone and high estrogens [25,26,27]. Additionally, the author assumed that a 2D:4D ratio referred to genes that influence the formation of limbs and the urogenital system [26]. Although we did not find a significant difference in the 2D:4D ratio between patients with Poland syndrome and healthy controls, we proved that the lengths of the second and fourth digits are statistically different in the group of women with Poland syndrome between the affected side and the nonaffected side and also between studied patients and healthy women. In patients with Poland syndrome, on the affected side, digits are underdeveloped and shorter in comparison with digits on the nonaffected side and with digits in the healthy control group. This may suggest that there is no advantage in prenatal testosterone level over estrogens, or vice versa, but their levels are proportionally reduced, which may cause underdevelopment of the fingers but no disturbances in the 2D:4D ratio. Nevertheless, further studies are needed to check this theory, especially in a group of men with Poland syndrome.

There are some reports paying attention to the fact that in patients with Poland syndrome, limb hypoplasia on the affected side can occur; however, the reported data are not detailed or current (from more than 40 years ago) [17,18]. In 1976, Ireland et al. mentioned that in PS patients, the arm and more frequently the forearm are hypoplastic [17]. In 1982, Senrui et al. noticed that abnormalities in Poland syndrome may include not only the bones but also soft tissues [18]. This information, of course, is important in describing the clinical picture of Poland syndrome, but it does not give precise values of disturbances. In our group, we showed that the length of the forearm on the affected side is statistically shorter than the length of the forearm on the nonaffected side, in patients with Poland syndrome (for more than 1 cm), and additionally, we proved that a similar significant difference is also present between dimensions of the forearm in patients with PS and those of healthy controls (more than 2 cm).

The literature mentions that Poland syndrome is more common in men than in women. It is totally the opposite in our sample: the majority of patients in the study group are women. This is probably because female patients more often consult a plastic surgeon to treat their breast asymmetry, which is due to the hypoplasia of the breast at the affected side. The majority of male patients are probably referred to orthopedic surgeons, which is why some cooperation between specialists is needed to increase the cohort of measured patients.

This study is not free from limitations. This is mainly due to the small number of examined cases. However, to our knowledge, this is the first study describing the digit ratio in patients with Poland syndrome. In the recent literature, the number of articles describing a large number of patients with Poland syndrome is rather small. In 2016, Vaccari et al. presented a group of 120 patients with Poland syndrome, but these authors investigated the prevalence of chromosomal imbalance and copy number variations and did not present a clinical picture of this syndrome [28]. In addition, in 2016, Wu et al. described the developmental characteristics of various types of hand bones from patients with Poland syndrome in a group of 32 patients. The authors performed X-rays on both hands, so their analysis was limited to only this part of upper limb [29]. Ten years ago, Catena et al. presented a new proposal for a hand and upper-limb-anomaly classification in patients with Poland syndrome [2]. Their group consisted of 175 patients with Poland syndrome who were divided into eight types of hand and upper-limb anomalies: the first type lacked the defect in the hand/upper limb; the second type had a hypoplastic hand; the next three types had different forms of symbrachydacytyly of the hand; the sixth type had a classic hand anomaly with radioulnar synostosis; the seventh type had aclassic hand anomaly with high scapula; and the eighth type had other associated anomalies [2]. The presented classification is based mainly on hand defects, and the only upper-limb malformation is synostosis. In our study, it is emphasized that not only is the hand significantly affected in Poland syndrome but so is the forearm, and its examination and/or measurements should always be performed when making a diagnosis of Poland syndrome.

## 5. Conclusions

In patients with Poland syndrome, the upper limb on the affected side is significantly different from the upper limb on the healthy side; it is seen mainly in the aspect of the length of the forearm and digits, which are shorter. Similar differences are also seen between patients and healthy controls, which strengthens the direction of described correlations. This can suggest that the whole upper limb on the affected side is underdeveloped, even when the hand seems to have grown normally (evident hand anomalies are not visible). When patients with Poland syndrome are examined, we think it is essential to pay attention not only to hand anomalies but also to the development of the whole upper limb. This recommendation may prove to be helpful in estimating the complete picture of Poland syndrome.

Although Poland syndrome is more frequent in men, our group is based on women. This might be associated with the profile of plastic surgery specialization and social awareness that female breast operations are in the scope of plastic surgeries. Case series reporting, especially by doctors of various specialties, would be essential to helping recognize the global incidence rate of Poland syndrome and determining whether the gender disparity is true.

## Figures and Tables

**Figure 1 jcm-11-07253-f001:**
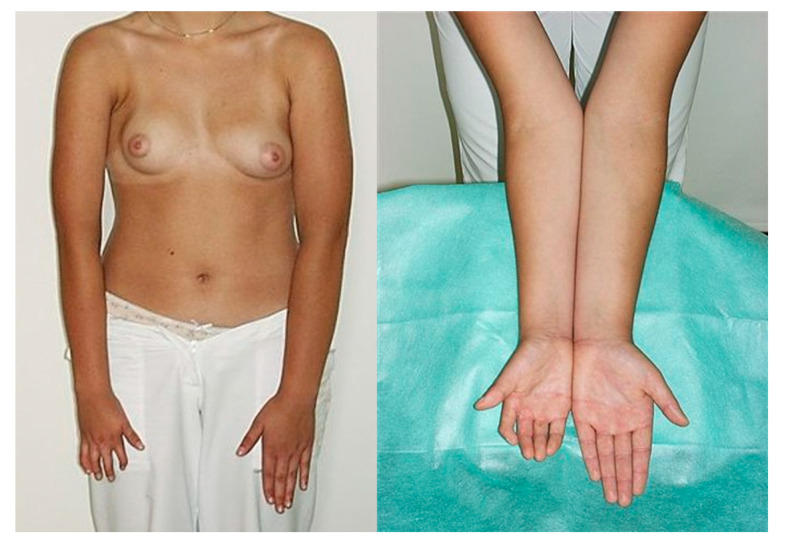
25-year-old female patient with right-sided Poland syndrome (frontal view of the chest after right breast reconstruction with silicone implant and hand appearance).

**Figure 2 jcm-11-07253-f002:**
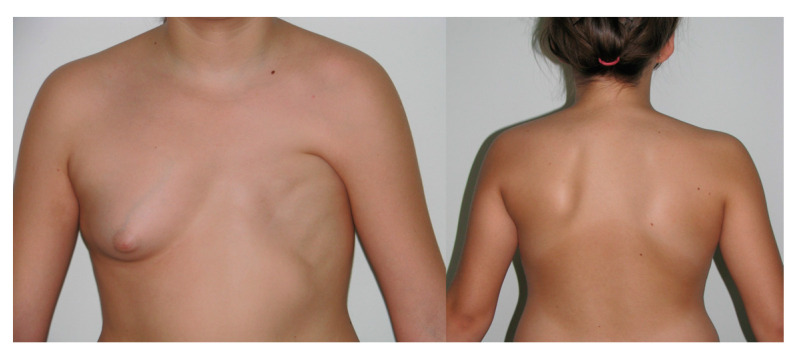
21-year-old female patient with left-sided Poland syndrome (frontal and back view).

**Figure 3 jcm-11-07253-f003:**
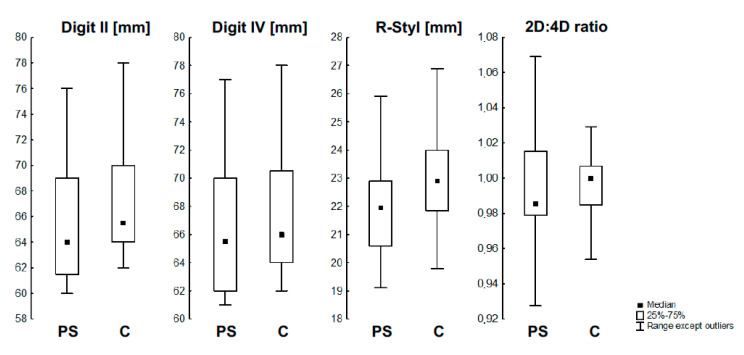
The distribution of measured parameters between groups.

**Table 1 jcm-11-07253-t001:** Comparison of hand parameters between affected and nonaffected sides in patients with Poland syndrome.

	Poland Syndrome				Poland Syndrome				*p*
	Affected Side *n* = 36				Nonaffected Side *n* = 36				
	Mean	SD	Min	Max	Mean	SD	Min	Max	
**Digit 2 length [mm]**	61.7	12.4	31.0	76.0	67.1	4.2	62.0	78.0	** 0.000 **
**Digit 4 length [mm]**	62.5	13.2	29.0	77.0	67.5	4.2	62.0	78.0	** 0.004 **
**2D:4D ratio**	0.99	0.05	0.88	1.14	0.99	0.03	0.91	1.06	0.197
**r-styr length [mm]**	22.0	1.9	19.1	25.9	23.1	1.7	19.8	26.9	** 0.000 **

**Table 2 jcm-11-07253-t002:** Comparison of hand parameters between affected side of patients with Poland anomaly and control group.

	Poland Syndrome Affected Side *n* = 36				Control Group *n* = 100				*p*
	Mean	SD	Min	Max	Mean	SD	Min	Max	
**Digit 2 length [mm]**	61.7	12.4	31.0	76.0	67.4	2.4	62.0	73.0	** 0.003 **
**Digit 4 length [mm]**	62.5	13.2	29.0	77.0	67.6	2.5	62.0	73.0	** 0.049 **
**2D:4D ratio**	0.99	0.05	0.88	1.14	1.00	0.02	0.94	1.03	0.208
**r-styr length [mm]**	22.0	1.9	19.1	25.9	24.7	1.6	22.0	28.0	** 0.000 **

## Data Availability

Not applicable.

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
