# Peer review of "Upper-Limb Disturbances in Female Patients with Poland Syndrome, including the Digit Ratio (2D:4D)"

_jcm, 2022, doi:10.3390/jcm11247253_

Round 1

Reviewer 1 Report

Thankyou for giving me the possibility to review this interesting paper, aiming to present disturbances in the development of upper limb in females with Poland syndrome (including digit ratio 2D:4D) in comparison to the healthy controls.

The paper is well written and deals with a very interesting topic. 

However, before considering it for pubblication, the following concerns should be addressed:

1. Please add two clinical cases images to the manuscript 

2. Please improve the discussion session (discussing further the relevance of the study and comparing your results to international litterature findings)

Author Response

Thank you for your positive feedback of our article.

We introduced all suggested remarks:

  1. We added clinical images of patients with Poland syndrome to our manuscript. Thank you for this valuable advice.
  2. The discussion section was rearranged and expanded. Hope it will meet your acceptance in this form. Comparison between our results and literature findings is quite difficult to perform as our description of digit ratio and forearm length in patients with Poland syndrome in comparison to healthy control is, to our knowledge, the first. However we added additional literature to present the strength of our research.

Reviewer 2 Report

Dear authors 

I think the study is minimal 

What you want to show is already known

I think you should find other correlations and study criteria within the group of patients studied.

Thank you

Author Response

Thank you for your comment. We agree with the reviewer that anthropometrical features of the Poland Syndrome have been extensively studied, however the 2d:4d ratio remains still not fully explored issue. The current study presents for authors best knowledge the largest sample regarding statistical association between Poland Syndrome, 2d:4d ratio and forearm length. Our study has proven that despite the anthropometric changes, the 2d:4d ratio remains the same for the affected side as for the healthy side – demonstrated in a precise manner and by use adequate power analysis. The line of development did not affect this pattern. Comparison between our results and literature findings is quite difficult to perform as our description of digit ratio and forearm length in patients with Poland syndrome in comparison to healthy control is, to our knowledge, the first. However we added additional literature to present the strength of our research. Hope in this form you will find our article more interesting.

Round 2

Reviewer 1 Report

The manuscript has been significantly improved in this revised version.

Author Response

Thank you for your positive opinion of our revised article. The figure 1 was changed as requested by the second Reviewer, as the basis of our analyzed group was female patients. Hope you will agree with that.  

Reviewer 2 Report

Dear authors, 

I think figure 1 should be changed: it represents the case of a 5 years old male patient and the study was done on adult female patients.

I think the conclusions should highlight the results of the study, the high incidence in female patients,...

Thank you

Author Response

Thank you for your comment.

The figure 1 was changed, you are right it is better to present only females cases.

The conclusions were changed, indicated issues were highlighted.

Hope current form of our manuscript will meet your acceptance.